# Molecular Dynamics Simulations Reveal the Conformational Transition of GH33 Sialidases

**DOI:** 10.3390/ijms24076830

**Published:** 2023-04-06

**Authors:** Xueting Cao, Xiao Yang, Min Xiao, Xukai Jiang

**Affiliations:** 1National Glycoengineering Research Center, Shandong Key Laboratory of Carbohydrate Chemistry and Glycobiology, NMPA Key Laboratory for Quality Research and Evaluation of Carbohydrate-Based Medicine, Shandong University, Qingdao 266237, China; 2State Key Laboratory of Microbial Technology, Shandong University, Qingdao 266237, China

**Keywords:** sialidase, GH33, HMOs, molecular dynamics simulations, catalytic mechanisms, protein engineering

## Abstract

Sialidases are increasingly used in the production of sialyloligosaccharides, a significant component of human milk oligosaccharides. Elucidating the catalytic mechanism of sialidases is critical for the rational design of better biocatalysts, thereby facilitating the industrial production of sialyloligosaccharides. Through comparative all-atom molecular dynamics simulations, we investigated the structural dynamics of sialidases in Glycoside Hydrolase family 33 (GH33). Interestingly, several sialidases displayed significant conformational transition and formed a new cleft in the simulations. The new cleft was adjacent to the innate active site of the enzyme, which serves to accommodate the glycosyl acceptor. Furthermore, the residues involved in the specific interactions with the substrate were evolutionarily conserved in the whole GH33 family, highlighting their key roles in the catalysis of GH33 sialidases. Our results enriched the catalytic mechanism of GH33 sialidases, with potential implications in the rational design of sialidases.

## 1. Introduction

Human milk oligosaccharides (HMOs) are the third most common component in human milk [1] and are believed to sharpen the gastrointestinal microbiota by hindering the adhesion of pathogens to the gut lining [2]. Sialyloligosaccharides account for 12.6–21.9% of the total free HMOs [3], which are particularly important for the neonate growth, including innate immunological, gastrointestinal maturation, and cognitive development [4,5,6,7]. As a novel class of prebiotics, the industrial production of sialyloligosaccharides has attracted considerable attention in recent years.

Sialyloligosaccharides are currently synthesized through either chemical or biological techniques. As for the chemical synthesis, multiple steps of protection and deprotection of the active moieties on sialic acid are necessary for the desired reaction [8,9]. In the biosynthesis, sialyltransferases and sialidases directly employ the sialic acid as the reactive substrate, and often display much improved regio- and stereo-selectivity compared to the chemical synthesis [10]. Sialyltransferases employ pricey cytidine 5′-monophospho-N-acetylneuraminic acid (CMP-Neu5Ac) as the glycosyl donor, significantly increasing the cost in the industrial production of sialyloligosaccharides [8,11]. Previous studies revealed that sialidases from *Arthrobacter aureus*, *Bifidobacterium infantis*, *Bacteroides fragilis* and *Clostridium perfringens* were able to synthesize sialyloligosaccharides [12]. Notably, these sialidases utilize cheap Casein glycomacropeptide (CGMP) or sialic acid oligomer as the glycosyl donor, instead of CMP-Neu5Ac used by sialyltransferases [13,14,15]. However, the conversion rates catalyzed by sialidases are often lower than 30% [12], attenuating their industrial applications. Despite great efforts, the empirical-based protein engineering of sialidases is often unsatisfied [16,17]. Understanding the structure, dynamics and function relationship of sialidases becomes necessary for the rational design of sialidases with improved transglycosylation activities.

To date, the sialidases that are known to show transglycosylation activity that predominantly stems from the GH33 family. The GH33 family consists of 9470 members and is mainly composed of sialidases from bacteria and simple eukaryotes [18]. The structure of GH33 sialidases is consisting of a catalytic domain and carbohydrate binding domains. The catalytic domain is featured by a six-blades fold, each of which contains a four-anti-parallel β-sheet (Appendix A) [19]. Sialidases catalyze trans-sialylation reactions via a classical ping-pong mechanism [18]. Among them, TcTS from *Trypanosoma cruzi* has received particular interest as a highly stereospecific trans-sialidase [20]. TrSA from *Trypanosoma rangeli* has 70% identity with TcTS, but is a strictly hydrolytic enzyme without transglycosylation activity [21]. Based on the sequence and structure of TcTS, a TrSA mutant with five mutations (TrSA5mut) showed weak trans-sialidase activity [21]. Both NanI from *Clostridium perfringens* str. 13 and RgNanH from *Ruminococcus gnavus* ATCC29149 have a typical fold of GH33 sialidases and are reported to show transglycosylation activity [14,19,22]. Notably, RgNanH is an intramolecular trans-sialidase that cleaves off the terminal α-2,3-linked sialic acid from glycoproteins, and produces 2,7-anhydro-Neu5Ac [23].

In the present study, we employed comparative molecular dynamics (MD) simulations to investigate the structural dynamics of GH33 sialidases. The overall structural features of four sialidases (TcTS, TrSA, NanI and RgNanH) and their interactions with the substrates were characterized in detail. We found that only the sialidases that possess transglycosylation activity formed a new cleft in the simulations. Molecular docking results revealed that the new cleft may serve to accommodate the glycosyl acceptor. Further, the key residues responsible for the interaction with the substrate were identified and found to be highly conserved in the GH33 family. Our results demonstrate that the opening of a new cleft plays a key role in the transglycosylation activity of GH33 sialidases.

## 2. Results

### 2.1. Overall Structure of GH33 Sialidases

The root mean square deviation (RMSD) of GH33 sialidases protein backbone atoms was calculated to evaluate whether the system reached the equilibrium state. All systems reach an equilibrium state after 50 ns. In the substrate-free system, the RMSD values of TrSA, TcTS, NanI and RgNanH converged around 0.29 nm, 0.19 nm, 0.14 nm and 0.27 nm, respectively (Figure 1). In the substrate-bound system, the RMSD values of TrSA, TcTS, NanI and RgNanH converged around 0.21 nm, 0.18 nm, 0.17 nm and 0.24 nm, respectively (Figure 1). Collectively, these results indicated that these systems were well equilibrated in the simulation and can be used for the subsequent analysis.

### 2.2. Structural Flexibility of GH33 Sialidases

To explore the structural flexibility of GH33 sialidases, we calculated the root mean square fluctuation (RMSF) for each enzyme in the substrate-free and substrate-bound states using the last 50 ns simulations. It was evident that the active center of the enzymes always showed very low flexibility, while the surrounding loops were relatively flexible. In the substrate-bound state, the flexible regions of TrSA included the loop from 21E to 27I, the loop from 142S to 148T and the loop from 165P to 173T (Figure 2a); for TcTS, the flexible regions included the loop from 21E to 27T and the loop from 141S to 148T (Figure 2b); the flexible regions of NanI included the loop from 286R to 293P, the loop from 364N to 420T, the loop from 520E to 523K and the loop from 628V to 636P (Figure 2c); and RgNanH included the loop from 279G to 284K, the loop from 321P to 330Q, the loop from 361G to 367A, the loop from 438N to 445L, the loop from 651G to 656D and the loop from 699S to 706K (Figure 2d). The conformational flexibility of the loops surrounding the active center may provide the necessary microenvironment for the catalysis. Notably, the flexible loops found in the NanI and RgNanH correspond exactly to the regions that open a second active site, as described in the following section.

### 2.3. The Conformational Transition of GH33 Sialidases

When we visualized the trajectory structures of the simulations of GH33 sialidases generated from molecular dynamics simulations, a new cleft formed in TcTS, NanI and RgNanH, but not in TrSA (Figure 3). Importantly, the scenario was reproducible in the simulation replicas (Appendix A). In TcTS, a new cleft formed due to the motion of the loop from 310G to 315D (Figure 2b and Figure 3b). In NanI, the formation of the new cleft primarily resulted from the movement of the loop from 286R to 293P (Figure 2c and Figure 3c). In RgNanH, the loop from 279G to 284K was driven to the open of the new cleft (Figure 2d and Figure 3d).

Next, although both TcTS, NanI and RgNanH displayed transglycosylation activity, only TcTS employed the β-galactoside as the glycosyl acceptor, while NanI employed the methanol as the acceptor, and RgNanH is an intramolecular trans-sialidase [14,20,23]. Therefore, TcTS was a better representative of the enzymes with transglycosylation activity. Therefore, we employed the rigid molecular docking for TcTS to investigate the potential role of the new cleft. We found that the binding free energy of the glycosyl acceptor (i.e., lactose) with the innate active site was −0.86 kcal/mol, while it was −1.35 kcal/mol with the new cleft (Appendix A). This suggested that the lactose preferred to bind to the new cleft. Notably, it has been reported that TcTS, NanI and RgNanH displayed transglycosylation activity [14,19,20], while TrSA did not [21]. This was consistent with their abilities to open the new cleft. Taken together, these results revealed that the conformational transition probably mediates the binding of the glycosyl acceptor, and thereby plays an essential role in conferring the transglycosylation activity of GH33 sialidases.

### 2.4. Key Structural Moieties of the Substrate Recognized by GH33 Sialidases

In order to identify the key structural moieties of the substrate that are recognized by GH33 sialidases, we analyzed the number of hydrogen bonds between the enzyme and the individual functional groups of the substrate using the last 50 ns simulations. We found that the carboxyl groups of the sialic acid part of the substrate were always the key sites that were recognized by the enzymes. TrSA formed approximately 4 hydrogen bonds with Sia-1COOH, and 1 hydrogen bond with Sia-8OH, Sia-9OH, Gal-4OH, and Gal-6OH, respectively; very few hydrogen bonds formed with the hydroxyl groups (Glc-OH) on the glucose residue (Figure 4). TcTS showed a very similar interaction pattern with the substrate, except that more hydrogen bonds formed with the hydroxyl groups on sialic acid (Figure 4). For NanI and RgNanH, approximately 5 and 3 hydrogen bonds were formed with the Sia-1COOH group, respectively (Figure 4). The amino acid residues of enzymes that contributed to the hydrogen bonding interactions have been provided in Appendix A. These results suggested that GH33 sialidases preferred to interact with the sialic acid part (i.e., glycosyl donor) compared to the lactose part (i.e., glycosyl acceptor).

### 2.5. Key Residues Responsible for Interacting with the Substrates

To identify the key residues of GH33 sialidases responsible for the substrate binding, we calculated the interaction energy between the substrate and each active site residue using the last 50 ns simulations, and analyzed the conformational changes of key residues by comparing the MD structures with the PDB structure (Figure 5). We found that the electrostatic interaction played the determinant role in the binding to the substrate; and the conformations of the residues in the active site of enzymes changed accordingly to ensure such interactions. In TrSA, the electrostatic interactions from 36R, 54R, 97D, 231E, 246R and 315R, and the hydrophobic interaction from 313W dominated the binding to the substrate (Figure 5a). Interestingly, we found that upon binding to the substrate, the side chains of 36R, 97D, 231E, 246R and 315R rotated toward the Gal-4OH, Sia-5NH, Sia-9OH, Sia-8OH, and Sia-1COOH groups, respectively, improving the interaction with the substrate (Figure 5a). The aromatic ring of 313W rotated inwardly by about 6.5 Å, which mediated the π-π packing with the ring of glucose (Figure 5a). A similar trend was found in TcTS, NanI, and RgNanH (Figure 5b–d). In addition to the formation of the new cleft, these findings revealed that the residue-level conformational transitions of GH33 sialidases also played a role in the binding to the substrate and the following catalysis.

### 2.6. Conservation Analysis of GH33 Sialidases

We investigated the sequence conservation in the GH33 family to examine the evolutionary significance of the key residues. As expected, most residues in the active site were highly conserved in the whole GH33 family, while the residues in the remaining part were variable (Figure 6a). This clearly indicated the importance of the residues in the active site for the enzymatic catalysis. In particular, the top five key residues identified for GH33 sialidases (e.g., 35R, 53R, 96D, 230E, 245R and 314R in TcTS, Figure 6b) were always highly and even absolutely conserved in the whole GH33 family (Figure 6c). The residues that are located in the new cleft were found to be relatively variable, compared to those in the innate active site. The evolutionary analysis revealed that the sequence evolution in the region of the new cleft may lead to the different transglycosylation activity of GH33 sialidases, despite an overall similar binding mode between the substrate and different GH33 sialidases.

## 3. Discussion

Human milk oligosaccharides are important nutrition for neonatal growth [2,24]. Sialyloligosaccharides are significant components of HMOs, which may improve intestinal maturation and cognitive development in neonate [25]. Although sialidases offer a promising strategy to synthesize sialyloligosaccharides, the poor activity often hindered their industrial applications [24]. Meanwhile, the knowledge gap of the catalytic mechanism of sialidases significantly limited their rational design to improve their transglycosylation activity [16,17,26]. Here, we investigated the structural dynamics of GH33 sialidases using comparative all-atom molecular dynamics simulations. We discovered that during the simulation of GH33 sialidase, conformational rearrangement of the active site occurred, and a new cleft formed to accommodate the acceptor. These results provide novel insights into the catalytic mechanism of GH33 sialidases, and a potential application for the rational design of sialidases to improve their transglycosylation activity.

The most interesting finding was the conformational transition in TcTS, NanI and RgNanH. Previous studies of TcTS reported a similar phenomenon, where a new cleft formed to recognize the β-galactoside as the glycosyl acceptor [27,28]. Our simulations revealed that the unique conformational transition also occurred in other GH33 sialidases (i.e., NanI and RgNanH). Notably, these two sialidases have been found to display transglycosylation activity [14,19]. By contrast, TrSA did not form the new cleft in our simulations, and correspondingly did not show transglycosylation activity based on the literature [21]. Therefore, the conformational rearrangement of GH33 sialidases plays a key role in their transglycosylation reaction through creating a new cavity to accommodate the glycosyl acceptor.

Another important finding was the conformational transitions of the residues in the enzyme active site. Previous studies of TcTS reported that the arginine triad (i.e., 35R, 245R, 314R) preserved the carboxylic group of sialic acid in the catalysis; 230E stabilized the enzyme-substrate intermediate; and 53R, 96D and 120W also aided in the stabilization of the sialic acid [29,30,31,32]. Based on our simulation results, all of these reported key residues adjusted their conformations to either favor the interactions with the sialic acid, or mediate the following opening of the new cleft, consistent with previous experiment-based assumptions [33,34,35]. Importantly, these functional residues were found to be highly conserved in the whole GH33 family, which implied that the conformational arrangements of GH33 sialidases, including both the opening of the new cleft and the residue-level adjustment, would be a general rule of their transglycosylation activity.

To the best of our knowledge, this is the first all-atom study to uncover the systematic conformational dynamics of GH33 sialidases. Excitingly, our simulation results disclosed the new cleft of GH33 sialidases, which was largely dependent on the conformational flexibility of enzymes. The unique conformational transition creates a new cavity to accommodate the glycosyl acceptor and thereby facilitates transglycosylation reaction. These findings provide a new clue to improve the transglycosylation activity of GH33 sialidases by rationally engineering the essential, new cleft.

## 4. Methods

### 4.1. Protein Preparation

A data collection comprising homologous sialidases from GH33 family was created for this study. Four sialidases were selected including TrSA (PDB: 1MZ5) from *Trypanosoma rangeli*, TcTS (PDB: 1S0I) from *Trypanosoma cruzi*, NanI (PDB: 2BF6) from *Clostridium perfringens* str. 13, and RgNanH (PDB: 4X47) from *Ruminococcus gnavus* ATCC29149. All protein structures were obtained from the Protein Data Bank (https://www.rcsb.org, accessed on 2 April 2023). The functional residues of these sialidases have been listed in Appendix A. Notably, TcTS, NanI and RgNanH showed transglycosylation activity, but TrSA did not [14,19,20,21]. Their evolutionary locations are scattered in the phylogenetic tree of the GH33 family (Appendix A). The structural similarity and functional variation made them good candidates for examining the structure–catalysis relationship of GH33 sialidases.

### 4.2. Molecular Dynamics Simulations

All MD simulations were performed using the GROMACS 2019 program with the CHARMM36 all-atom force field. The protein was solvated using the SPC model in a cubic box with the dimensions of 8 × 8 × 8 nm^3^ [36]. To produce a neutral system with 0.15 mol L^−1^ ionic concentration, appropriate amounts of Na^+^ and Cl^−^ were added by replacing water molecules randomly [37]. In order to eliminate steric interference, steepest energy minimization was performed with 5000 steps for every system to give the maximum force below 1000 kJ mol^−1^ nm^−2^ [38]. Then, the system was firstly equilibrated for 200 ps under the isothermal–isochoric (NVT) ensemble and another 200 ps under the isothermal–isobaric (NPT) ensemble [39]. Complete equilibration was assessed by the convergence of the potential energy and the temperatures of the systems. Lastly, 100 ns production MD simulations were performed with three replicas in the NPT ensembles [40]. V-rescale and Parrinello–Rahman methods were used to control the system temperature (310K) and pressure (1 bar), respectively [41,42]. The LINCS algorithm was used to constrain all bonds to hydrogen atoms in the protein and the SETTLE algorithm was used for the water molecules [43,44]. The Particle Mesh Ewald (PME) method was used to evaluate the long-range electrostatic interactions [45]. The non-bonded pair list cutoff was 10.0 Å and the pair list was updated every 10 fs.

For each sialidase studied here, two MD systems were prepared, including the protein-only system and protein–substrate complex system. The detailed information of these simulation systems has been provided in Table 1.

### 4.3. Data Analysis

The biophysical properties of GH33 sialidases were analyzed using the internal tools in GROMACS. A hydrogen bond (gmx_hbond) was defined if the acceptor–donor distance was less than 0.35 nm and the acceptor–hydrogen–donor angle was less than 30° [46]. The RMSD (gmx_rms) and RMSF (gmx_rmsf) were determined to examine the structural stability and flexibility after eliminating the overall translational and rotational movements by superimposing the C_α_ atoms of each snapshot structure onto the initial structure using least-squares fitting [47,48]. The interaction energy (gmx_energy) between amino acid residues and substrate was calculated based on the CHARMM36 all-atom force field [49]. Data-analysis-related formulas were provided in the Appendix A. The structural visualization was performed using Polo [50].

Docking of the lactose into the binding sites of TcTS was performed using the AutoDock4.0 program. Docking files were prepared using AutoDock Tools 1.5.6 software [51]. The detailed information of these molecular docking systems has been provided in Appendix A. MEGA 11 was used to conduct the evolutionary study of the GH33 family. The CONSURF SERVER (https://consurf.tau.ac.il, accessed on 2 April 2023) was used to examine the conservation of a specific sequence position in GH33 family. The Weblogo (https://weblogo.berkeley.edu, accessed on 2 April 2023) was used to generate the sequence profile of the GH33 sialidases active site as described in the literature [52].

## Figures and Tables

**Figure 1 ijms-24-06830-f001:**
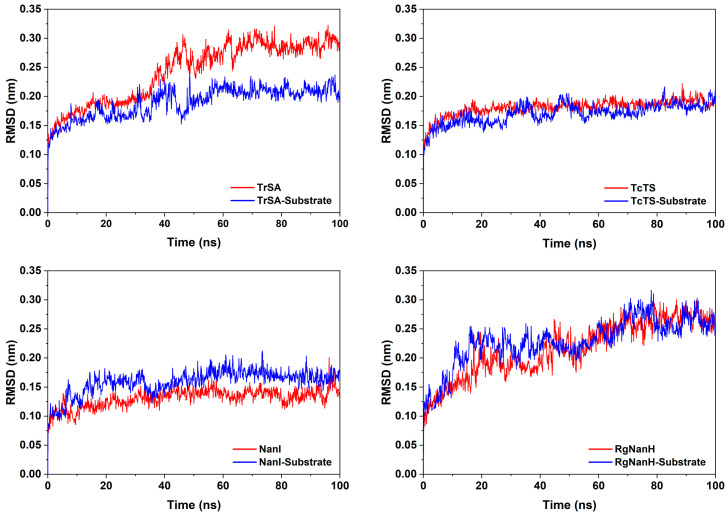
Time evolution of the backbone RMSD for GH33 sialidases with and without the substrates. The calculated data of the substrate-free and substrate-bound states are shown in red and blue, respectively.

**Figure 2 ijms-24-06830-f002:**
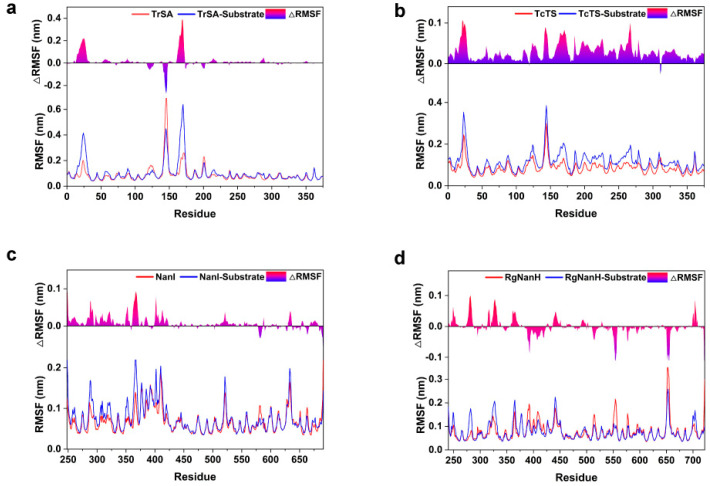
Structural flexibility of GH33 sialidases with and without the substrates. The RMSF values of the substrate-free and substrate-bound states are shown in red and blue lines, respectively. The changes in the RMSF are shown with filling bars. Structural flexibility of TrSA, TcTS, NanI and RgNanH with and without the substrates are shown in (**a**–**d**), respectively.

**Figure 3 ijms-24-06830-f003:**
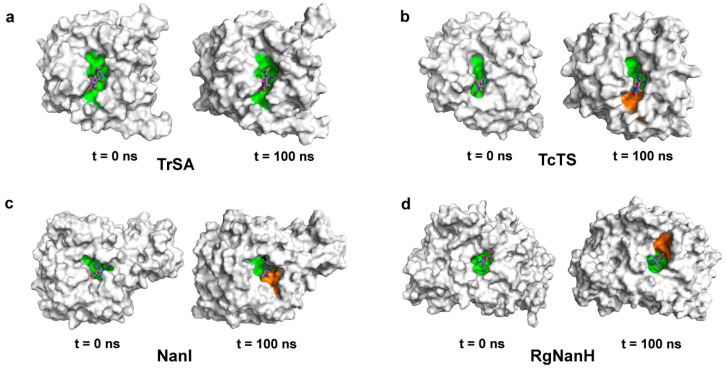
The conformational transition of GH33 sialidases in MD simulations. The proteins are shown in surface models, the innate active site and the new cleft of the enzyme are highlighted in green and orange, respectively. The conformational transition of TrSA, TcTS, NanI and RgNanH are shown in (**a**–**d**), respectively.

**Figure 4 ijms-24-06830-f004:**
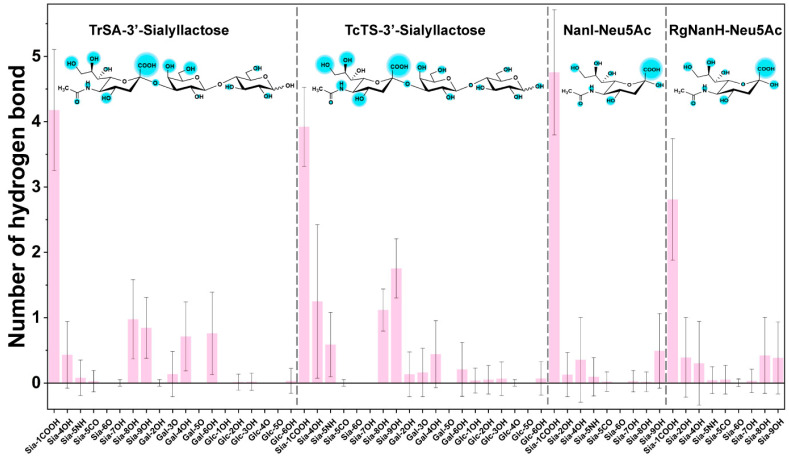
Hydrogen bonding interaction between the substrate and GH33 sialidases. The hydrogen bonds were analyzed based on the MD simulations. The sizes of the cyan cycles shown in the chemical structures of the substrate represent the strength of the hydrogen bonding interactions. Sia-1COOH, the carboxyl group linking to C1 of sialic acid; Sia-4OH, the hydroxyl group linking to C4 of sialic acid; Sia-5NH, the acetyl amino group linking to C5 of sialic acid; Sia-5CO, the acetyl amino group linking to C5 of sialic acid; Gal-2OH, the hydroxyl group linking to C2 of galactose; and Glc-1OH, the hydroxyl group linking to C1 of glucose.

**Figure 5 ijms-24-06830-f005:**
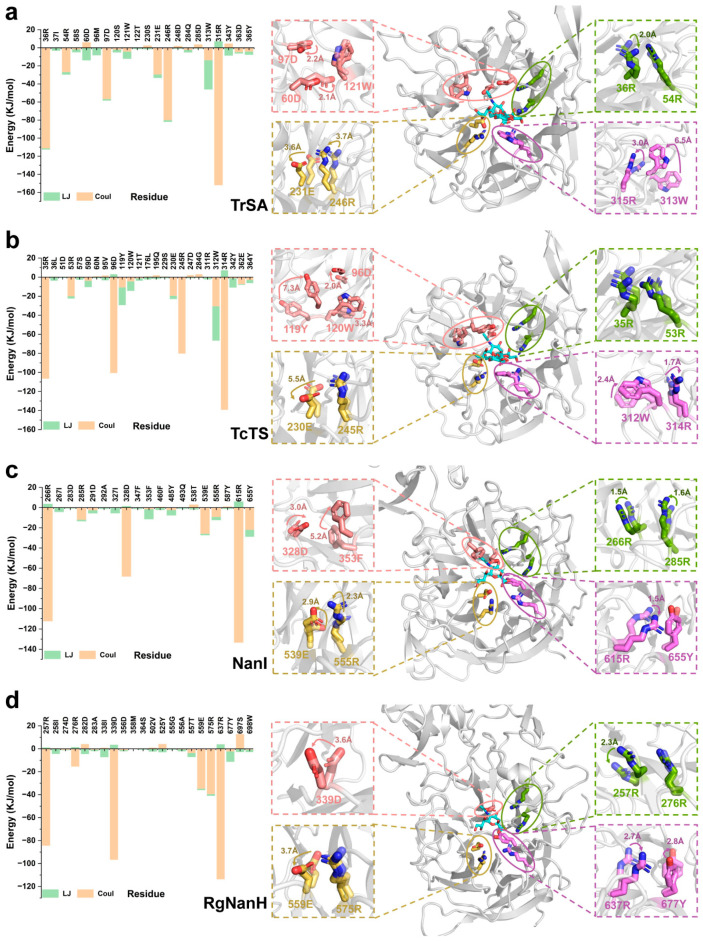
Interaction energy between the substrate and GH33 sialidases and the conformational transitions of the active site residues. The conformational changes of key residues are analyzed by comparing the MD structures with the PDB structure. LJ: hydrophobic interactions; Coul: electrostatic interactions. The orientation directions of the residues are shown with arrows and the changed distances are labelled. Interaction energy between the substrate and TrSA, TcTS, NanI and RgNanH and the conformational transitions of the active site residues are shown in (**a**–**d**), respectively.

**Figure 6 ijms-24-06830-f006:**
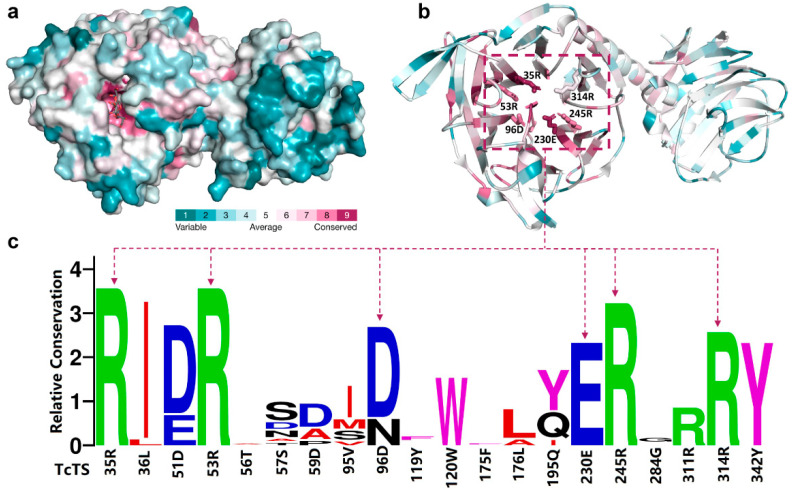
Conserved analysis of GH33 sialidases. (**a**) Conservation analysis of GH33 sialidases. The gradient color spectrum from green to red represents the increase in conservation (**b**) The key functional residues in TcTS. (**c**) Active site sequence profiles of GH33 sialidases. The ordinate indicates the relative degree of conservation, while the abscissa represents amino acid ID of TcTS. Each type of amino acid is represented by abbreviated letters with a corresponding color (KRH: green; DE: blue; MVALI: red; FWY: purple; Others: black), where the same color suggests a similar property of the amino acid. In each row of sequence profiles, the heights of all letters represent the relative degrees of conservation, while the height of a single letter denotes its specific occurrence frequency.

**Table 1 ijms-24-06830-t001:** Detailed information of molecular dynamics simulation systems.

System	PDB ID	Number of Water	Number of Na^+^	Number of Cl^−^
TrSA	1MZ5	51,135	145	150
TrSA-3′-Sialyllactose		51,178	145	146
TcTS	1S0I	47,222	133	139
TcTS-3′-Sialyllactose		47,212	133	138
NanI	2BF6	27,279	89	77
NanI-Neu5Ac		27,283	90	77
RgNanH	4X47	23,393	86	66
RgNanH-Neu5Ac		23,373	87	66

## Data Availability

The data presented in this study are available on request from the corresponding author.

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
