# Peer review of "Molecular Dynamics Simulations Reveal the Conformational Transition of GH33 Sialidases"

_ijms, 2023, doi:10.3390/ijms24076830_

Round 1

Reviewer 1 Report (New Reviewer)

In the present study, authors have performed MD simulations of GH33 sialidases, and have claimed to find a new active site based on simulations. Please find my comments below,

 1) Firstly, it is crucial to note that the claim of discovering a new active site based on simulations alone is not entirely reliable without experimental validation. Also, since the authors have performed a single trajectory simulation, the sampling may not be sufficient to make any definitive claims. Instead of calling it an active site, it would be more accurate to refer to it as an allosteric site.

2) Moreover, the authors have not provided sufficient structural information about the enzymes in question. In the introduction section, it would be essential to mention the structures of all four enzymes and provide figures focusing on the active site region.

3)  Additionally, it would be helpful to include a brief discussion of previous studies done on these enzymes in the introduction.

Author Response

  1. Firstly, it is crucial to note that the claim of discovering a new active site based on simulations alone is not entirely reliable without experimental validation. Also, since the authors have performed a single trajectory simulation, the sampling may not be sufficient to make any definitive claims. Instead of calling it an active site, it would be more accurate to refer to it as an allosteric site.

Response: We thank the reviewer for the suggestions. To validate the simulation predictions, we conducted two simulation replicas for each system and analyzed their structural changes in the simulations. We found that both TcTS, NanI and RgNanH always experienced significant conformational transitions in the simulation replicas, while TrSA did not. As you suggested, we changed the phrase and used “conformational transition” to describe this phenomenon in the revised manuscript.

  1. Moreover, the authors have not provided sufficient structural information about the enzymes in question. In the introduction section, it would be essential to mention the structures of all four enzymes and provide figures focusing on the active site region.

Response: We thank the reviewer for the suggestion. In the revised manuscript, we have added more information of these GH33 sialidases, and provided a new figure to highlight their active sites (Figure S1).

  1. Additionally, it would be helpful to include a brief discussion of previous studies done on these enzymes in the introduction.

Response: We thank the reviewer for the suggestion. We have added several sentences to introduce the previous studies of these enzymes in the revised manuscript.

Reviewer 2 Report (New Reviewer)

The paper present conventional calculation methods to investigate the dynamics and binding possibilities of ligands interacting with sialidases. It is an interesting subject but the level of details is not informative enough to help the reader to understand, validate, and reproduce the calculations.  

No clear details are provided about the nature of the calculations (except in Table 1 for the MD systems). A Table should also be given for molecular docking systems. By the way, in Table 1, is it really necessary to show the columns entitled “temperature” and “pressure”? As the values are always the same, specifying them in the Table caption should be enough.

Figure S3 reports only one system studied by docking calculations. Why not the other systems (and a discussion in the main text)?

Also, the origin of the results is not systematically provided, e.g., when discussing the results, but also in Figure and Table captions, e.g., in Table S1.

The authors mention the use of MD triplicates, but they do not specify whether their analyses (RMSD, RMSF, Figure 4, …) are carried out over one of three triplicates. Also, they do not give any details regarding the similarities between the replicates.

Regarding the RMSD profiles (Figure 1), there is no information about the reference system (PDB or optimized or …) that is used to calculate the RMSD values.

Also, the authors mention that the MD production stages are carried out for 100 ns (section 4.2). At the beginning of section 2.1, they mention that an equilibrium state is obtained after 50 ns. Does that mean that further analyses are carried out over the last 50 ns of the MD trajectories (it should be the case – averages are to be calculated over equilibrated parts of the MD trajectories)?

No clear information is provided to help the reader to understand how the second active site is detected. The authors say “visually”. What does that mean? Figure 3 displays models with the innate site (green) and the second site (orange). The reader who is not accustomed to the biological systems under study wonder why is the “long” and apparently accessible site in TrSA is not composed of two parts (green and orange) like in the other proteins? Also, in Figure 3, why are there two pictures for each protein? The answer is not given in the Figure caption. The structures in Figure 2 do not help much because they provide static representations too.

In Figure 2, the loop associated with the second active site is characterized by large RMSF values in NanI and RgNanh. But not in TcTS. Why?

Where do the results shown in Figure 4 (H bonds) come from? MD? Molecular docking? Is there consistency between both approaches?

Section 2.5: the authors found that some residue side chains are rotated upon ligand binding; I assume they compare the MD structures with the PDB structure. But it is not mentioned in the text.

The MD simulations are carried out with the CHARMM36 force field (see section 4.2). However, the interaction energy calculations use the Amber force filed (as mentioned in section 4.3). Why such a change (using Gromacs, it should be possible to get interaction energies using the same force field as the MD one)?. How is it related to reference 46 mentioned in the text?

Section 4.3 present some information about the docking calculations, but it is uncomplete. Is it a semi-flexible approach or a rigid one? If semi-flexible, what are the degrees of freedom? How many poses were considered for the analysis? Only the best one? What is the link between the score function and the binding energies?

Author Response

The paper present conventional calculation methods to investigate the dynamics and binding possibilities of ligands interacting with sialidases. It is an interesting subject but the level of details is not informative enough to help the reader to understand, validate, and reproduce the calculations.

Response: We thank the reviewer for the comments. We have revised the manuscript to clarify the study points according to your following constructive suggestions.

  1. No clear details are provided about the nature of the calculations (except in Table 1 for the MD systems). A Table should also be given for molecular docking systems. By the way, in Table 1, is it really necessary to show the columns entitled “temperature” and “pressure”? As the values are always the same, specifying them in the Table caption should be enough.

Response: We thank the reviewer for the suggestions. (1) We have provided more details of moecular dynamics simulations in the revised manuscript. (2) We have provided the method of molecular docking in the Supporting Information (Table S3). (3) We have changed the Table 1 as you suggested.

  1. Figure S3 reports only one system studied by docking calculations. Why not the other systems (and a discussion in the main text)?

Response: We thank the reviewer for the comments. Although both TcTS, NanI and RgNanH displayed transglycosylation activity, only TcTS employed the β-galactoside as the glycosyl acceptor, while NanI employed the methanol as the acceptor, and RgNanH is an intramolecular trans-sialidase. Therefore, TcTS seems to be a better representative of traditional sialidases with transglycosylation activity. We have discussed this point in the revised manuscript.

  1.  Also, the origin of the results is not systematically provided, e.g., when discussing the results, but also in Figure and Table captions, e.g., in Table S1.

Response: We thank the reviewer for the suggestions. We have revised the manuscript to make the results more clearly.

  1. The authors mention the use of MD triplicates, but they do not specify whether their analyses (RMSD, RMSF, Figure 4, …) are carried out over one of three triplicates. Also, they do not give any details regarding the similarities between the replicates.

Response: We thank the reviewer for the comments. For each system, we performed three simulation replicas to examine the reproducibility of the results. In particular, we focused on the structural dynamics of the studied enzymes. Notably, TcTS, NanI and RgNanH always experienced significant conformational transition in three simulation replicas. We have clarified this in the revised manuscript.

  1. Regarding the RMSD profiles (Figure 1), there is no information about the reference system (PDB or optimized or …) that is used to calculate the RMSD values.

Response: We thank the reviewer for the comment. The enzyme structure from the first frame of MD trajectory was used as the reference system to analyze the RMSD. We have added this information in the revised manuscript.

  1.  Also, the authors mention that the MD production stages are carried out for 100 ns (section 4.2). At the beginning of section 2.1, they mention that an equilibrium state is obtained after 50 ns. Does that mean that further analyses are carried out over the last 50 ns of the MD trajectories (it should be the case – averages are to be calculated over equilibrated parts of the MD trajectories)?

Response: Yes, most analysis were conducted based on the last 50-ns simulations. We have explained this in the revised manuscript.

  1.  No clear information is provided to help the reader to understand how the second active site is detected. The authors say “visually”. What does that mean? Figure 3 displays models with the innate site (green) and the second site (orange). The reader who is not accustomed to the biological systems under study wonder why is the “long” and apparently accessible site in TrSA is not composed of two parts (green and orange) like in the other proteins? Also, in Figure 3, why are there two pictures for each protein? The answer is not given in the Figure caption. The structures in Figure 2 do not help much because they provide static representations too.

Response: We thank the reviewer for the suggestions. (1) In the whole simulations, TrSA always kept such a “long” active site, while TcTS, NanI and RgNanH experienced significant conformational transitions. To show the structural changes, the new opened clefts in TcTS, NanI and RgNanH were coloured in orange. (2) Two pictures of an individual enzyme represent the structural snapshots before and after the MD simulations, respectively. We think that the comparative views help the understanding of the conformational transition of these enzymes. (3) As you suggested, we have removed the cartoon models in this figure.

  1. In Figure 2, the loop associated with the second active site is characterized by large RMSF values in NanI and RgNanH. But not in TcTS. Why?

Response: We thank the reviewer for the comments. In NanI and RgNanH, the formation of their second active clefts were dependent on the flexible motion of loops, while it was not in TcTS. This indicated that GH33 sialidases may adopt different mechanisms to achieve the conformational transition, and the following transglycosylation activity.

  1. Where do the results shown in Figure 4 (H bonds) come from? MD? Molecular docking? Is there consistency between both approaches?

Response: We thank the reviewer for the comments. The hydrogen bonds were analyzed based on the MD simulations to examine the binding pattern of GH33 sialidases to their substrates. We did not use molecular docking to do such a similar analysis. We have clarified this in the revised manuscript.

  1. Section 2.5: the authors found that some residue side chains are rotated upon ligand binding; I assume they compare the MD structures with the PDB structure. But it is not mentioned in the text.

Response: Yes. We have added this information in the revised manuscript.

  1.  The MD simulations are carried out with the CHARMM36 force field (see section 4.2). However, the interaction energy calculations use the Amber force filed (as mentioned in section 4.3). Why such a change (using Gromacs, it should be possible to get interaction energies using the same force field as the MD one)?. How is it related to reference 46 mentioned in the text?

Response: We thank the reviewer for pointing out this mistake. Yes, we used CHARMM36 force field in the whole MD simulations, and the interaction energy was calculated by integrating gmx mdrun (with -rerun flag) and gmx energy commands. We have corrected this and updated the reference in the revised manuscript.

  1. Section 4.3 present some information about the docking calculations, but it is uncomplete. Is it a semi-flexible approach or a rigid one? If semi-flexible, what are the degrees of freedom? How many poses were considered for the analysis? Only the best one? What is the link between the score function and the binding energies?

Response: We thank the reviewer for the comments. We employed molecular docking to investigate the binding of the glycosyl acceptor (i.e. lactose) to the enzymes using a rigid approach. The molecular docking produced 50 binding complexes, and we selected the one with the best binding free energy. In such a complex, the lactose bound to the new-opened cleft of enzymes, which suggested that the new cleft found in the present study may serve to accommodate the glycosyl acceptor, and thereby play a key role in mediating the transglycosylation activity of GH33 sialidases. We have provided new information to clarify this point in the revised manuscript.

Reviewer 3 Report (New Reviewer)

The authors have proposed a molecular dynamics study of GH33 Silalidases. Overall, the manuscript is very well written. Here are some suggestions:

1. Why latest PDB entries were not selected for the MD simulation study?

2. Can authors justify the NPT vs NVT for their complex system?

3. Minimization steps should be described in the MD simulations protocol.

4. Please provide the formulas for all the required studies.

5. Did the authors have performed energy minimization of all the pdbs?

6. Why MM/GBSA calculations were not performed?

7. 

Author Response

  1. Why latest PDB entries were not selected for the MD simulation study?

Response: We thank the reviewer for the question. We selected these PDB structures because they contained the substrates in the active sites of enzymes, which allowed us to examine the interaction of enzymes with their substrates.

  1. Can authors justify the NPT vs NVT for their complex system?

Response: We thank the reviewer for the question. Previous studies revealed that the NPT system is suitable to study the structure and function of enzymes (Araki, M. et al., Nat Commun 2021, 12 (1): 2793; Lambert, E. et al., Nat Commun 2022, 13 (1): 1022). Therefore, we also employed the NPT system for the study.

  1. Minimization steps should be described in the MD simulations protocol.

Response: We thank the reviewer for the suggestion. We have added this information in the revised manuscript.

  1. Please provide the formulas for all the required studies.

Response: We thank the reviewer for the suggestion. We have provided the formulas in the Supporting Information.

  1. Did the authors have performed energy minimization of all the pdbs?

Response: We thank the reviewer for the question. Yes, we performed energy minimization and equilbration for all the structures of studied enzymes.

  1. Why MM/GBSA calculations were not performed?

Response: We thank the reviewer for the question. In the present study, we mainly explore the relationship between the conformational transition of GH33 sialidases and the transglycosylation activity. We employed interaction energy analysis to identify both the key amino acid residues of enzymes and the key structural groups in the substrates. Compared to the overall binding free energy generated from MM/GBSA, we think that these results were more suitable to explain the binding pattern of GH33 sialidases. We have provided several sentences to clarify this point in the revised manuscript.

Round 2

Reviewer 1 Report (New Reviewer)

The authors have made changes suggested by me. The manuscript can be accepted in the present form.

Author Response

The authors have made changes suggested by me. The manuscript can be accepted in the present form.

Response: We thank the reviewer for the great efforts to review the manuscript.

Reviewer 2 Report (New Reviewer)

The authors replied to the comments made during the first reviewing round. On a methodological point of view, the paper is not innovative, but I assume it can be seen as a starting point for further studies by interested readers.

Author Response

The authors replied to the comments made during the first reviewing round. On a methodological point of view, the paper is not innovative, but I assume it can be seen as a starting point for further studies by interested readers.

Response: We thank the reviewer for the great efforts to review the manuscript.

Reviewer 3 Report (New Reviewer)

The authors have not understood the questions and did not reply the comments appropriately.  

Author Response

The authors have not understood the questions and did not reply the comments appropriately. 

Response: We thank the reviewers for their great efforts to review our manuscript. According to their kind comments and suggestions, we have added more methodological details, performed new data analysis and changed the results description during the revision process. This helps significantly improve the quality of the study.

This manuscript is a resubmission of an earlier submission. The following is a list of the peer review reports and author responses from that submission.

Round 1

Reviewer 1 Report

Cao et al report in their manuscript the results of short (100 ns) MD simulations (without replicas) on 4 sialidases either in the presence and in the absence of their substrates, to claim evidence for the discovery of a second active site in these enzymes.

In my opinion, the data to support this claim are not sufficient, and the manuscript is not correctly structured, therefore in the present form is not acceptable for publication. Here a list of the concerns that I hope will help the authors to improve their manuscript for a possible future resubmission.

1) First of all, the manuscript is not correctly structured. Methods should be reported either before the Results section, or at the end of the manuscript, not between the Results and the Discussion. Moreover, Figure 1 must be the first figure that the readers see, not the last one!

2) Table S1 must be included in the main text, not in the Supporting material. Supporting information should report instead evidence about the fact that the MD simulations reached the stability.

3) The introduction is insufficient to explain the scientific problem. For example, it is not clear why the authors focused their research on GH33 sialidases. Moreover, they selected four members of this family of sialidases without explaining why did they selected exactly these members and not others. Considering also that the authors talk diffusely about the discovery of a "new active site" of these enzymes, a detailed explanation of the enzymatic mechanism and structural features of these enzymes is mandatory.

4) Results: The authors claim they have discovered a "second active site" of the enzymes, never identifying it exactly from a structural point of view, nor explaining the functional role of this putative "active site" in the mechanism of action of the protein. Moreover, the evidence for the presence of this "active site" is simply a very small perturbation of RMSF (less than 0.1 nm) in two out of 4 systems (NanI and RgNanH) in a 100 ns long simulation made without replicas. For TcTS, despite the claim of the authors, I can't see any difference in the flexibility of the region highlighted in Figure 3b. Moreover, the supposed enhanced flexibility in TcTS is in a zone of the protein very different from that of the other two enzymes. In the same figure 2B, I see a more evident (although hardly significant) RMSF difference in the segments 21-27 and 141-148, which are not discussed by the authors, with an apparently evident bias. Additionally, the authors support their claim with figure 4 showing a barely detectable change in the surface charge potential of the protein which I would like to understand how it is correlated with the supposed presence of the "second active site". The sentence "This indicated that the binding of GH33 sialidases to the sialic acid might drive the formation of the second active site in order to accommodate the lactose" is highly speculative and must be supported by other, more consistent evidences. Paragraph 2.6 should also show and discuss the conservation of the supposed "second active site" to corroborate this finding. Are the residues affected by the (supposed) significant RMSF difference conserved across the different species or not?

5) Methods: the authors should justify why they selected these starting structures for their simulations, also in terms of quality of the structures (are there other structures of these enzymes available? If yes, what is their quality?). MD simulation setup must be described in more details (for example, ref. 27 is unappropriate to describe how the authors neutralized the system). In Data analysis, authors state "The interaction energy (gmx_energy) between amino acid residues and substrate was calculated based on the amber force field equation". To the best of my knowledge, gmx_energy is not an "interaction energy", but the command to extract energy components from an energy file. So the authors must detail how did they calculate the "interaction energy" between the enzyme and the substrate.

4) In my opinion, the entire discussion is highly speculative, as the evidences reported are not sufficiently solid to claim the discovery of a "second active site", which moreover is never clearly identified in the structures.

Other detailed concerns:

Caption of Figure 4 is misleading, as the figures do not show the "formation" of the putative second active site, but only the surface charge potential (that the authors claim to be different in the two figures of each panel). In my opinion, this small difference could also be related to the fact that the orientation of the structures in each panel is not exactly the same. By the way, the authors should identify the structure bound to the ligand and the structure without ligands inside, and should explain what are these structures (are they the output .gro file? Are they a representative structure from a cluster? Which cluster?)

Figure 5: no Hydrogen bonding interactions are reported in figure 5, since the residues involved in these interactions are not shown in this figure.

Figure 7: the caption is misleading. Moreover, the "sequence profile" is only a partial profile made on selected residues (with unknown criteria), and it is useless to support the claim of the presence of a "second active site".

Reviewer 2 Report

This manuscript uses molecular dynamics studies to gain more insight into substrate binding effects on GH33 sialidase family proteins protein conformation and identify conserved amino acids which contribute to opening of a second active site. 

As written this manuscript is hard to follow and should be modified to increase clarity which will help increase impact:

- The introduction should report how many members in the GH33 protein family. 

- Much of the information in the first paragraph of section 2.1 including Fig. 1 should be moved to the Introduction. None of this information including protein name, source organism or activity is currently present. It is confusing when the acronyms for these four sialidases are mentioned in the results with no preface. The substrate that these enzymes are crystallized with should also be given. The substrates are only first mentioned in Fig. 5. This information should be given earlier and in Table. S1.

- The number of amino acids in each protein should be given. Definition of the domain boundaries of the catalytic and inactive central domain would be also be helpful. This will allow the reader to understand where the residues referred to in later sections reside.

- A table of experimentally-determined catalytic residues would be helpful for comparison purposes. 

- There is no description of the overall sialidase structure depicted in Figure 1. Did the authors model this? 

- Figure 2 - More descriptive figure title is needed.

- Figure 4 - A ribbon diagram rather then surface representation will better provide the reader of an understanding of any structural similarities in the second active site. The dashed line box should be another color. It is hard to see in the figure.
